# Study of Metakaolinite Geopolymeric Mortar with Plastic Waste Replacing the Sand: Effects on the Mechanical Properties, Microstructure, and Efflorescence

**DOI:** 10.3390/ma15238626

**Published:** 2022-12-02

**Authors:** Ivana Soares, Francisco X. Nobre, Raimundo Vasconcelos, Miguel A. Ramírez

**Affiliations:** 1School of Engineering and Sciences Guaratinguetá, São Paulo State University—UNESP, Av. Ariberto Pereira Cunha, 333, Guaratinguetá 12516-410, SP, Brazil; 2Faculty of Technology, Federal University of Amazonas, Av. General Rodrigo Otávio, 1200, Coroado I, Manaus 69067-005, AM, Brazil; 3Federal Institute of Education, Science and Technology of Amazonas, Av. Sete de Setembro, n°. 1975, Manaus 69020-120, AM, Brazil

**Keywords:** geopolymer, sustainability, plastic waste, metakaolinite, efflorescence

## Abstract

In this study, the production of a mortar was proposed in which plastic waste replaced sand by 0%, 50%, and 100% in order to create a sustainable alternative for construction. The performance of the mortars was tested with two types of activators, one with NaOH, as a simple activator, and the other with NaOH and Na_2_SiO_3_, as a compound activator. The effects of the LDPE plastic bag waste and the activators on compressive strength, porosity, microstructure analysis, and efflorescence formation were correlated and discussed. The results showed that the replacement of sand with plastic waste at 50% and 100% proportionally reduced the compressive strength due to the increase in porosity caused by the waste, especially in the group of mortars with the simple activator, and included the formation of efflorescence. On the other hand, the compound activator increased the packing of the particles in the mortar, as observed in the images of the microstructure. This reduced porosity inhibited efflorescence and resulted in higher resistances that reached a maximum value of 22.68 MPa at 28 days in the group of 50% mortars with the compound activator. Therefore, the study showed that there is potential for the replacement of sand with plastic waste for the production of mortars, which can be considered a more sustainable building material.

## 1. Introduction

Geopolymers produced from metakaolinite have demonstrated good results regarding compressive strength and have the potential to be used as sustainable building materials [1,2]. Metakaolinite is one of the most commonly used raw materials in geopolymerization and has emerged as an alternative to mitigate the environmental impacts of CO_2_.

Metakaolinite is obtained via the calcination of kaolin, which has high levels of the mineral kaolinite and occurs at a moderate temperature between 650 and 800 °C. This alternative, when compared to the production of Portland cement, contributes to reducing the amount of CO_2_ released into the atmosphere [3]. According to McLellan et al. [4], geopolymers produced without external heating show a reduction of 44–64% of gases released into the environment when compared to Portland cement.

Given that the annual production of cement in 2015 was 4.3 billion metric tons and that there is an estimated growth of 6.1 million metric tons for 2051 [5], there is a need to seek sustainable alternative materials that reduce the consumption of common cement [6,7], and one of the alternatives is to use geopolymers.

Geopolymers are versatile materials from the point of view of synthesis. The raw material with the presence of aluminosilicates can be activated with alkaline solutions of sodium hydroxide, silicate, and sodium [3,8], as well as with acidic solutions of phosphoric acid, according to new research by Nergis et al. [9], who obtained results similar to the synthesis with alkalis.

Although it is environmentally very important to reduce the consumption of Portland cement, material innovation can be a great opportunity to integrate environmental and social issues in terms of the destination of solid waste. Among these wastes, plastic stands out for being a very frequently consumed material that, at the end of its useful life, has proven to be a major environmental problem. The global production of plastics for 2030 is estimated at 550 million metric tons, which implies that there is a trend toward a large increase in the disposal of plastic waste that will end up often in the oceans or in the ground [10]. Balwada et al. [11] state that plastics are a cause for great concern, given that their destination, in the vast majority of cases, is landfill sites.

Moreover, waste incineration as a solution becomes another problem from the point of view of CO_2_ emissions into the atmosphere [5,12]. It is noted that the generation of plastic waste in post-consumption may generate a greater environmental impact than the current one. Among the most used plastics, low-density polyethylene (LDPE) stands out. It is a polymer with a density of between 0.91 and 0.94 g/cm^3^ and is very resistant at temperatures below 60 °C. It is a material that is widely used in the production of plastic bags and beverage and food packaging, among other uses [13].

These advantages highlight the reason for the large volume of LDPE used in the world, which contributes to the increase in waste generation. Consequently, the management and disposal of plastic waste from post-consumer activities has become a global challenge for the environment [14]. It should be noted that the disposal of plastic waste is a major cause of environmental impacts and has become a major problem due to its very low biodegradability. In view of this, the search for alternative ways to reuse these wastes has resulted in important research, such as the use of plastic waste as an aggregate for the production of mortars and concretes made with Portland cement [15,16,17,18].

Therefore, recycling and utilization, in addition to extending the useful life of the material, also add environmental value. However, when it comes to geopolymer-based concretes and mortars, very little research focuses on the use of recycled plastics [14,19]. Thus, there is a gap in studies on the use of plastic waste, especially regarding the use of LDPE, which is one of the most commonly discarded plastics in the environment due to its high consumption and reduced useful life. In addition, the use of plastic waste as a component in construction materials has become a sustainable alternative [20].

Like any other material, geopolymers are also susceptible to the formation of pathological manifestations [21]. Efflorescence is one of the most important problems that occurs during the use of geopolymers because, in addition to causing aesthetic damage to the surface and the physical properties, it can cause deep degradation of the concrete and corrosion of the reinforcement, which compromises the durability and safety of the structure. Thus, some studies have been developed to identify the factors of this phenomenon, such as alkaline concentrations [22].

Efflorescence occurs through the combination of leaching and carbonation processes when the unreacted alkalis in the geopolymerization migrate to the surface of the geopolymer by leaching and then evaporating leaving behind the salts. Thus, the result is the formation of a precipitate of sodium carbonate when using a sodium hydroxide activator (NaOH) or potassium carbonate from a potassium hydroxide (KOH) activator [23]. The degree of porosity of a material is a potential factor for efflorescence formation since the porous structure is related to the leaching of alkalis and their mobility [24].

As such, the scientific contribution of the research in question is the proposal of an alternative material to Portland cement in the production of a mortar that can be applied in construction. The use of plastic waste in building materials has the advantage of being more sustainable, reduces CO_2_ emissions into the atmosphere, and can help mitigate the problem of plastic waste accumulating in the ocean and landfills. As a result of this context, the research presented in this work sought to study the replacement of sand with LDPE plastic waste from plastic bags in the production of a geopolymer mortar. The raw material used as a source of aluminosilicates was metakaolinite, which was obtained from clay in the metropolitan area of Manaus, Amazonas, Brazil.

Thus, the study began with the calcination of the kaolin to obtain metakaolinite. The mortars produced were influenced by activators and the replacement of the sand with plastic waste, which decreased the resistance and increased the porosity. This porosity contributed to the formation of efflorescence in the mortars activated with sodium hydroxide; however, the mortars activated with sodium hydroxide and sodium silicate, in addition to not forming the efflorescence, resulted in better resistances and lower porosity.

## 2. Materials and Methods

### 2.1. Raw Materials and Naming of the Samples

The kaolin used in this research was extracted in the municipality of Presidente Figueiredo, located in the metropolitan region of Manaus, Amazonas state, Brazil. The activators used in the synthesis of the geopolymer were NaOH, which is one of the most commonly used due to the reduced cost when compared to potassium hydroxide. NaOH was purchased from Nox Solutions in the form of flakes with 99% purity. The other activator was sodium silicate (Na_2_SiO_3_) from Quimidrol, Santa Catarina, Brazil, with the silica modulus (SM) provided by the manufacturer at the molar ratio SiO_2_/Na_2_O = 2.24 (Na_2_O = 14.5%, SiO_2_ = 31.5%) by mass. The sand used was riverbed sand and was acquired from suppliers in Manaus, Brazil. The LDPE plastic waste was obtained from the reciclagem Amazonas plant in the form of flakes, which represents the penultimate step in the recycling process before extrusion to form the grains. LDPE is collected in the form of packaging, such as plastic bags, that have already been discarded and have become waste. The collection is carried out in the city of Manaus, and the material goes through a cleaning and separation process. The washing removed dirt, and the separation graded the materials for density. After washing, the LDPE plastic waste was passed through a knife shredder and then through 8 mm, and 5 mm mesh sieves. In this study, LDPE plastic waste is referred to as plastic waste.

Thus, the geopolymeric (GP) mortars produced in the research were named according to the level of replacement of sand by plastic waste and according to the type of activator used, and are as follows: GP0-S, GP50-S, GP100-S and GP0-C, GP50-C and GP100-C, with 0%, 50%, and 100% being the percentages of replacement of sand by plastic waste. The letter S was adopted for the group of mortars with a simple activator of NaOH and the letter C for the group with the NaOH and Na_2_SiO_3_ compound activator.

### 2.2. Processing and Characterization of Kaolin

The processing of the kaolin was in accordance with the following steps: 1. Drying of the clumps for 7 days at room temperature, followed by manual crushing; 2. Then, the material was washed in a sieve (Tyler 200) with a mesh opening of 0.075 mm. This procedure helped in the removal of impurities and the fraction of sand present in the kaolin; 3. The kaolin that had settled after washing was then dried in an oven for 24 h at 100 °C to remove moisture and was left to cool in the oven and again manually crushed; 4. The kaolin processing was completed with dry sieving through a 0.075 mm sieve (Tyler 200) to control the particle size without the need for grinding.

After processing, the characteristics of the kaolin were identified in relation to the mineralogical phases by using X-ray diffractometry and its quantification with refinement using the Rietveld method and the GSAs-Expgui program. The XRD analysis was performed using an X-ray diffractometer (Panalytical, Empyrean, Malvern, UK) with Cu-Kα (λ = 0.1541838 nm) in an angular range of 10–80° of 2θ° with step size and time per step of 0.02° of 2θ and 60 s, respectively.

Thermogravimetry (DTG) and differential thermal analysis (DTA) were performed to characterize the nature of the kaolin in the heating. The equipment used was a simultaneous thermal analyzer (TA Instruments, SDT-Q600, Milford, MA, USA) using an inert atmosphere (N_2_), maximum flow of 30 mL/min, temperature range from 25 °C to 800 °C, and a heating rate of 10 °C.

Based on the results of the thermogravimetry, the kaolin was subjected to heat treatment by calcination at 650 °C for 1 h to produce metakaolinite. Accordingly, the material was placed in an electric muffle furnace (LINN, Elektro Therm, Győr, Hungary). The calcination process was started at room temperature until it reached 650 °C, and after calcination, the metakaolinite obtained was cooled for 12 h inside the muffle furnace and, finally, stored in resealable bags.

### 2.3. Characterization of Metakaolinite and Aggregates

To discover the chemical composition of metakaolinite and its long-range structure and find out whether the material has the potential for the production of a geopolymer, the metakaolinite was characterized using X-ray fluorescence (XRF) and, using stoichiometry, the ratio between the aluminosilicates was established. The analysis was performed in a spectrometer (Malvern Panalytical, EPSILON 3 XL, Malvern, UK), with a maximum voltage of 50 kV, a maximum current of 3 mA, and helium gas (pressure 10 atm/10 kgf/cm^2^).

X-ray diffractometry (XRD) measurements were used to determine the mineralogical phases of metakaolinite after heat treatment, and the conditions of this analysis were the same as those used for the kaolin. The microstructure was analyzed using a scanning electron microscope (Zeiss, EVO LS 15, Jena, Germany).

Granulometric analysis was also performed in order to obtain the particle size and specific surface area of the metakaolinite and the aggregates. The test was performed using laser scattering in a particle size analyzer (Malvern Instruments Mastersizer 200), with a rotation speed of 2200 rpm and time of 12 s, and a background of 20 s.

To establish the dosage of material used in the production of the mortars, the specific mass was obtained via helium gas pycnometry (Quantachrome Instruments, MPV P-6DC, Boynton Beach, FL, USA). The mortar dosage was based on the parameters suggested by Davidovits [25], i.e., 10 < H_2_O/Na_2_O < 25 and 0.22 < Na_2_O/SiO_2_ < 0.48. Thus, the stoichiometric calculation of the chemical compounds was applied to reach an ideal amount of simple and compound activator solution considering the ratio of water to metakaolinite (H_2_O/MK = 0.54), according to the study by Lahoti et al. [26], and 12 mol·L^−1^ for the NaOH solution.

When measuring the replacement of sand by plastic waste, the calculation took into account the difference between the specific masses of these materials. Table 1 shows the dosage required to produce the geopolymer mortars. Mortars of the GP-S and GP-C groups of geopolymers are named with the letter S for the simple NaOH activator and C for the compound activator with NaOH and Na_2_SiO_3_. The numbers 0, 50, and 100 refer to the percentage replacement levels of sand with plastic waste.

### 2.4. Manufacture of the Mortars

First, the activators were prepared, and the NaOH solution was obtained by mixing the solids at 12 mol·L^−1^ with distilled water. This solution stood for 24 h due to the exothermic reaction after homogenization. The compound activator was prepared by mixing the NaOH solution at 12 mol·L^−1^ with the Na_2_SiO_3_ solution and standing for 24 h.

The manufacture of the mortars began by manually mixing the dry materials for 3 min. First, the aggregates and then the metakaolinite, in order to obtain a better homogeneity between the particles. Then, the activator solution was added to the dry mixture, which was mixed again manually for 1 min. Once this was conducted, it was placed in a mechanical mortar mixer (MA-TEST, E 094X) at a speed of 140 rpm for 3 min so that the activator enveloped all the dry material, and then for another 3 min at a speed of 285 rpm in order to achieve a good homogeneity of the fresh mixture. The total mixing time was 7 min. The mixing time and speed were adapted from the studies of Longhi et al. [27].

After mixing, the mortar was poured into metal cylindrical molds of 50 mm in diameter and 100 mm in height and compacted for 60 s with the aid of a sieve shaker (VIATEST 220 V) to eliminate air bubbles. Then, the samples were covered in plastic film and remained in the mold for 24 h to cure at room temperature.

After removal from the molds, the samples were wrapped in plastic film and placed in resealable plastic bags to preserve the moisture of the material until the curing ages of 14 and 28 days. A total of 72 samples were produced, 12 for each mortar divided into 6 for each curing age. Other samples were also made in a silicone mold in the dimensions of 20 × 20 × 20 cm; 12 for the efflorescence test and 24 for the absorption and porosity tests. The Si–Al ratio in the geopolymers was 1.9 for GP-S and 2.4 for GP-C.

### 2.5. Characterization of Mortars

#### 2.5.1. Mechanical Strength Test

The compressive strength test was performed with a 200-ton load cell (EMIC PC 200 CS) and an axial load speed of 0.2 MPa/s, according to NBR 7215 [28]. This test was performed at the ages of 14 and 28 days and served to elucidate how much the replacement of sand with the waste and activators affected the mechanical performance of the mortars. Neoprene was used in the upper and lower part of the specimens in order to obtain a regularized surface for better distribution of stresses during the test, and the calculation of stresses was performed using Equation (1).
(1)$$\delta =\frac{F}{A}$$
where *δ* is the stress (MPa), *F* is the force (N), and *A* is the area (mm^2^).

#### 2.5.2. Porosity and Water Absorption Test

The porosity and water absorption tests were performed as described in the Brazilian standard NBR 9778 [29]. First, the dry mass was measured at air temperature and in an oven at a temperature of 105 °C for 24, 48, and 72 h. Once this was conducted, the oven-dried sample was immersed in water starting with 1/3 of its volume in the first 4 subsequent hours and then fully immersed in the remaining 64 h. Thus, the immersed mass was determined after 24, 48, and 72 h of immersion. In addition, the saturated mass immersed in a hydrostatic bath was measured. The calculation of porosity and void index was obtained using Equations (2) and (3).
(2)$$\mathrm{Porosity}=\frac{\mathrm{Msat}-\mathrm{Ms}\;}{\mathrm{Ms}}\times 100$$
(3)$$\mathrm{Void}\;\mathrm{ratio}=\frac{\mathrm{Msat}-\mathrm{Ms}\;}{\mathrm{Msat}.i-M}\times 100$$
where
Msat is the mass of saturated material;Ms is the mass of the oven-dried material;Msat.i is the mass of the immersed saturated material, and M is the mass of the material after curing.All masses were measured in grams.

#### 2.5.3. X-ray Diffractometry and Microstructure

X-ray diffractometry (XRD) measurements were performed on the mortars of the group GP-S and GP-C and, thus, determined the phases found after the synthesis of the geopolymer. The test conditions were the same as those used in the characterization of the kaolin.

The microstructure was analyzed using a scanning electron microscope (Zeiss EVO LS 15) and aided in the analysis of particle interaction and porosity.

#### 2.5.4. Efflorescence Test

The formation of efflorescence in the geopolymeric mortars was observed by performing a test according to the method described by Longhi et al. [27]. In this method, after curing, some of the samples were placed in ambient conditions, and the others were submerged in 50 mL of distilled water with drying and wetting cycles for 7, 14, and 28 days. The cycle consisted of removing the samples from the bath every day, wiping them with absorbent paper, and then returning to the immersion bath.

The samples that remained in ambient conditions were monitored for the formation of efflorescence under these conditions at the ages of 7, 14, and 28 days. As for the samples that were submerged, these were first removed from the water after 28 days, and the formation of efflorescence was also observed visually at the same ages.

The efflorescence crystals were collected on the surface of the mortars using the curettage method and were characterized using X-ray diffractometry and SEM, using the same test conditions as the kaolin characterization.

## 3. Results

### 3.1. Thermogravimetry of the Kaolin

Figure 1 shows the graph of the TGA and DTA curves of the kaolin. It was possible to observe that the first mass loss occurred at 55 °C, corresponding to 0.41%, which is related to water that is adsorbed on the outer surface of the particles. The second mass loss was 12.70% and occurred in the range of 456 °C to 652 °C, which can be attributed to the loss of organic material followed by the loss of structural water up to 652 °C.

The endothermic peak observed in Figure 1b, which occurred at 525 °C, corresponds to kaolin dehydroxylation, which is a result of the elimination of the hydroxyl group (OH) and occurs during the transformation process of kaolin into metakaolinite [30,31].

### 3.2. X-ray Diffractometry of Kaolin and Metakaolinite

Figure 2a shows the X-ray diffraction (XRD) pattern of the kaolin. The indexation reveals that there are three phases: dialuminium phyllo-disilicate tetrahydroxide—Al_2_(Si_2_O_5_)(OH)_4_ (kaolin) is the main phase, which has a triclinic structure with a space group of P1(1), lattice parameters a = 5.1577(15) Å, b = 8.9417(23) Å and c = 7.3967(40) Å, angles α = 91.672(50)°, β = 104.860(2)°, and γ = 89.989(2)° and a unit cell volume of 329.57 Å^3^. All diffraction peaks are in accordance with the crystallographic information available in Inorganic Crystals Structure Database (ICSD) card No. 68,968 and the literature. The second phase was indexed with ICSD card No. 90,145 for silicon oxide (SiO_2_) and exhibits a trigonal structure (P3221), lattice parameters of a = b = 4.9100 Å, c = 5.400 Å, and unit cell volume of 112.74 Å^3^. Finally, the third phase is the titanium oxide and anatase (TiO_2_) with a tetragonal structure a = b = 3.7892(4) Å, and c = 9.537(1) Å, and unit cell volume of 136.93(3) Å^3^.

From the XRD pattern shown in Figure 2a for kaolinite, the Hinchley index (HI) was calculated using the following Equation (4):(4)$$\mathrm{HI}=\frac{A+B}{A_{t}}$$
where A and B are the height of the diffraction peak at 2θ = 20.2° for ($\bar{1}10)$ plan and 2θ = 21.3° for (11$\bar{1}$), respectively, while the At is the height from the baseline to the peak. According to Liu et al. [32], an HI of ≥1.3 indicates a highly ordered structure, an HI of 1.1–1.3 is characteristic of an ordered structure, an HI of 0.8–1.1 is relatively disordered, and an HI of <0.8 is disordered. However, several factors should be considered in the analysis of these results, such as the presence of structural defects, dopants, mix of the phase with other minerals, and the amount of water molecules in the layers of aluminum and silicon clusters [33]. The calculated HI for kaolin in this study was 1.44, which indicates a highly ordered structure.

The phase composition was obtained by structural refinement using the Rietveld method, as can be seen in Figure 2b. As expected, via the Rietveld refinement, the occurrence of 90.71% of kaolinite, 6.18% of quartz, and 3.11% for anatase in the kaolin sample [14,34] was confirmed. This result is in accordance with what is reported by Faqir et al. [35], in which the use of natural kaolin clay was studied in order to obtain geopolymer by a reaction involving the natural kaolin, silica sand, sodium hydroxide, and water. The basal distance (d) of kaolin was calculated using the full width at half maximum (FWHM) of (001) plane at 2θ = 12.32° is 0.71 nm, in agreement with the basal distance reported by Tan et al. [36].

Based on the literature [37,38], TGA and DTA analyses, and the XRD patterns for metakaolinite in Figure 2c, it is suggested that the increase in the temperature in the thermal treatment of kaolin induces the total conversion of kaolin to metakaolinite at a temperature of 650 °C. The characteristic diffraction peak of kaolin at 2θ = 12.32° completely disappears at this temperature. On the other hand, the planes at 2θ = 20.8° and 2θ = 25.3°, attributed to the silicon oxide and anatase, respectively, show an increase in the intensity of their diffraction peaks. In addition, there was the occurrence of an amorphous region at 2θ between 15° and 30°, which is strong evidence for the obtention of metakaolinite due to the highly disordered structure of the kaolin at 650 °C. Thus, we suggest the following steps based on the literature [33,38].
$${\mathrm{Al}}_{2}({\mathrm{Si}}_{2}O_{5})(\mathrm{OH}{)}_{4}\;\overset{\sim 650\;{}^{\circ}C}{\to }\;{\mathrm{Al}}_{2}{\mathrm{Si}}_{2}O_{7}+2H_{2}O$$

The obtention of metakaolinite was confirmed by XRD analysis and structural Rietveld refinement, as shown in Figure 2c, for kaolin treated at 650 °C. Based on these results, the presence of 0.93%, 19.82%, and 79.25% for anatase, quartz, and disordered metakaolinite, respectively, was confirmed. In addition, It can be seen that the main intensity peak at 2θ = 12.32° disappears, which indicates the conversion of kaolinite to metakaolinite.

### 3.3. Chemical Composition of Metakaolinite Using XRF

Table 2 shows the chemical composition of the metakaolinite calcined at 650 °C for 1 h when analyzed using XRF. In this result, it was observed that the percentage of silica and alumina oxides totaled 92.21%, discounting the 6.18% of silica related to quartz (non-reactive material). Thus, it was possible to calculate the SiO_2_/Al_2_O_3_ ratio of the aluminosilicates, which resulted in 2.20. The silica modulus (SM = SiO_2_/Na_2_O) of the simple activator was equal to 0 since the simple solution is composed only of NaOH. In the compound activator group, the SM was 0.94, since in this group, in the stoichiometry, the SiO_2_ present in the sodium silicate solution was considered.

Thus, it was observed that the metakaolinite produced in the present work has more SiO_2_ in its composition and a reduced Fe_2_O_3_ and TiO_2_ content. This benefits the reactivity of the raw material in view of the high content of aluminosilicates and low content of other oxides that do not react in the solution to produce geopolymers.

### 3.4. Morphology of the Particles of the Raw Materials

In Figure 3, the micrographs of the kaolin before and after the heat treatment are shown. In the microstructure of the kaolin (Figure 3a), the stacking of alternating layers of SiO_2_ tetrahedra and Al_2_(OH)_6_ octahedra [39] of the kaolinite was observed. This morphology was destroyed, as seen in Figure 3b.

The larger kaolinite particles in Figure 3a became fragmented, as shown in Figure 3b (with red arrows), thus changing the initial morphology. This favored the phase transition and transformed the kaolin into metakaolinite, a more reactive material [40], which occurs because of the tetrahedral and octahedral coordination, and which is an electrically neutral crystalline structure and is broken down in the heat treatment.

It can be seen in Figure 3c that sand is a porous material (circles in red), unlike plastic waste (Figure 3d), which is a material without porosity, and this is characteristic of a hydrophobic and more homogeneous material.

### 3.5. Laser Granulometry of the Raw Materials

Table 3 shows the size distribution of the particles of metakaolinite, sand, and plastic waste (LDPE). The particle size of the metakaolinite in 10% (d10) is less than 2 µm, indicating that it is clay soil, according to the NBR 6502 standard [41]. The existence of larger particles in 50% (d50), 60% (d60), and 90% (d90) may be related to the lack of grinding of the kaolin in the processing, which is confirmed by the low specific surface area (1.38 m^2^/g).

In the sand, the granulometry in 5%, 60%, and 90% was within the range for fine sand and, in 10%, was classed as medium sand. The fineness modulus was 2.61, which was within the optimal zone variation of 2.2–2.9 for a fine aggregate, as established by NBR 7211 [42]. In view of the fact that plastic waste is not a natural aggregate, the particle size range was compared with sand. Thus, the granulometry of the LPDE in 10%, 50%, and 90% was compatible with medium sand. For 60%, it was not possible to relate the granulometry to any type of clay or sandy material.

Figure 4 shows the overlap graph of the curves of the metakaolinite, sand, and plastic waste by which the granulometric difference between the materials was observed.

### 3.6. Compressive Strength Test

Figure 5 shows the graph of the compressive strength for the mortars of the GP-S (Figure 5a) and GP-C (Figure 5b) groups at the ages of 14 and 28 days.

The two groups showed that, in the mean of the resistances, the results decreased proportionally with the increase in the replacement of the sand by plastic waste until reaching a reduction of approximately 16% in GP50-S and 100% in GP100-S at 28 days, when compared to the GP0-S reference mortar.

The increase in the concentration of alkalis can improve the solubility of aluminosilicates by increasing the concentration of Na_2_O, thus causing greater compressive strength [34]. As such, it can be inferred that the result of the resistance of the mortars GP-0, GP50-C, and GP100-C was favored by the increase in the concentration of Na_2_O via the combination of NaOH and Na_2_SiO_3_, in addition to the addition of silica in the Na_2_SiO_3_ solution. Thus, when comparing the two groups of mortar, it was noted that both the simple activators and the compound activators exerted a strong influence on the resistance.

In the end, it was noted that the best result was the GP-C in all dosages when compared to the GP-S mortars. The mortar with the best mechanical performance was GP50-C with a compressive strength of 22.69 MPa, which is a value that is very close to the average strength value found in the geopolymers produced in the study by Ahmadi et al. [43] who added carbon nanotubes. The nanotubes, also used in the synthesis of composites [44], bridge the particles and improve the mechanical properties. It is understood that in group C mortars, the compound activator may have acted as a binder between the particles, and this increased the resistance. However, the best mechanical performance in group C was the mortar GP50-C. It can be deduced that this mortar has the potential to be used in construction applications in materials that do not require high strength.

The Si–Al ratio defines the type of tridimensional structure formed in the geopolymerization and its applications in materials [45]. The Si–Al ratio in the GP-S mortar was 1.9, and for the GP-C, it was 2.4. It can be inferred that, for group C mortars, the high Si–Al ratio contributed to the higher dissolution of the aluminosilicates.

### 3.7. Absorption and Porosity Test

The absorption and porosity graph in Figure 6 shows an increasing trend of absorption and porosity in the mortars after 28 days, in accordance with the increase in the replacement of sand by plastic waste in each formulation. However, in the GP-S group, the porosity was more expressive when compared to the GP-C group. The compound activator may have influenced the decrease in the porosity of the group GP-C (Figure 6b) when Na_2_SiO_3_ was added, which may have contributed to improving the adhesion of the aggregate particles and the matrix, thus resulting in higher compressive strengths, as observed in item 3.6.

According to Bai and Colombo [1], porosity and strength have been investigated by using several models proposed by many authors. This is due to the fact that they represent a significant defect in the strength of geopolymers. This was observed in this study when comparing porosity with compressive strength. The dosage parameters and the raw material are practically the same, but the difference in the type of activator was sufficient to alter the behavior of the matrix, as discussed in item 3.9.

### 3.8. Phase Identification in the GP-S and GP-C Mortars Using XRD

The diffractograms of the GP-S group mortars in Figure 7 show the spectra with crystalline peaks that correspond to the same atomic ordering of quartz and anatase observed in the XRD of metakaolinite, as similarly observed in the studies of Azevedo et al. [45] and Bature et al. [46]. These phases do not contribute to the formation of polymer chains. Other quartz peaks were observed as a function of the sand used in the mortars. These peaks were reduced as the plastic waste content increased with the replacement of the sand.

The formation of a new phase, zeolite A, was observed, which, according to Longhi et al. [27] and Wan et al. [47], in geopolymers is relatively common in a matrix of metakaolinite activated with hydroxides due to the pseudozeolytic structure. The zeolitic phase was also observed in the studies of Bature et al. [46] in the manufacture of a NaOH-activated mortar. In zeolites, porosity may contribute to decreased strength due to its microporous crystalline structure [48,49], as observed in the compressive strength test.

The amorphous band, which was initially from 15° at 30 °C in the metakaolinite, underwent a shift to 18° at 38 °C. This displacement, according to Azevedo et al. [45], is related to the formation of the aluminosilicate gel (N-A-S-H).

According to Davidovits [50], three-dimensional structures have an atomic ratio of Si–Al that varies from 1 to 3, and the lower it is, the more it is possible to obtain zeolitic three-dimensional structures formed by the Si-O-Al-O bond, poly(sialate) that come from a structural reorganization that occurs in the synthesis of the geopolymer. In the field of materials applications, poly(sialate) can be used in the production of bricks and ceramics and as a fire protection material.

Figure 8 presents the XRD of the GP-C mortars, which shows a wide and diffuse halo at angles of 18° at 38 °C, and which is associated with the structural disorder, in addition to defined peaks corresponding to quartz and anatase already present in kaolin and few peaks of zeolite. Thus, it can be deduced that in this research, two types of geopolymer mortar were obtained, namely, the GP-S group with the highest degree of crystallinity and the GP-C group with the highest crystalline disorder and the lowest amount from the zeolite phase.

The N-A-S-H gel of the GP-C group did not show a predominance of the zeolitic phase, so it can be inferred that the three-dimensional network with the new atomic rearrangement formed a Si-O-Al-O-Si-O bond, poly(sialate-siloxo). In the field of materials applications, this type of geopolymer can be used in the production of cement and concretes with low CO_2_ emissions, as well as encapsulation of reactive toxic waste, since it results in more resistant materials [50].

The difference in the bonds of poly(sialate) and poly(sialate-siloxo) gel is in the bonding of silicon oxides (O-Si-O) that occurs in geopolymerization, either through the silicate-based activator solution or through the addition of silica fume.

### 3.9. Microstructure of the Mortars

Figure 9 shows the microstructure of the mortars that were obtained from the specimens after the compression test. The existence of fissures resulting from compression fracture and also from the fragmentation of the samples for SEM analysis can be observed.

All the mortars of the GP-S group showed the formation of a very porous composite. This porosity can be attributed to the granulometry of the particles due to the lack of packing, which hindered the interaction between the matrix and the plastic waste. Another factor to be considered is that the plastic waste because it is not porous, nullified the anchoring effect with the matrix, while in the sand, this effect was more significant. In addition, the GP-S group has much more unreacted metakaolinite in the synthesis (Figure 9(a2)) in relation to the GP-C group. This may be associated with the NaOH activator that was not sufficient to dissolve the aluminosilicates.

When comparing GP-S to GP-C, it is noted that the result was a more compact matrix that favored pore reduction and which contributed to an increase in the resistance. It can be inferred that the compound activator assisted in the interaction between the matrix particles and the aggregate when it formed the three-dimensional Si-O-Al-O-Si-O network, thus resulting in better packing when adding Na_2_SiO_3_ to the NaOH solution.

In both the groups of mortars, the metakaolinite did not react fully in the synthesis, and this may be related to the low specific surface area. According to the studies by Weng et al. [51], the larger specific area of metakaolinite is responsible for accelerating the setting time and increasing the strength, i.e., the larger specific surface area of metakaolinite is related to its reactivity potential. However, this did not significantly interfere with a reduction in resistance in this study. The mortar EDS is shown in Appendix A.

### 3.10. Efflorescence Test

Figure 10 shows the formation of efflorescence in the GP-S group and GP-C group mortars when exposed to the environment after curing.

Efflorescence formation was observed in the GP-S group in samples “a” (exposed to the environment) at the ages of 7, 14, and 28 days, while in samples “b” (submerged), after 7 days, the water was migrating to the surface, gradually at 14 and 28 days, bringing with it the crystals, which crystallized and presented a similar behavior to the “a” samples. In the GP-S group, all the mortars aggressively developed efflorescence, which transported the composite material and degraded it, as observed in a more expressive form in Figure 10a,b at 28 days.

According to Tam et al. [52], alkalis and water are factors that control efflorescence, as the water that is attracted to the pores by capillary suction evaporates to the surface, transporting alkaline ions and causing efflorescence.

However, the whitish product formed on the surface of the samples cannot be observed visually. This may have been due to the amount of transported material that mixed with the salts. The sodium leachate formed in the samples may indicate insufficient geopolymerization [53], and this was observed in the results of the microstructure of the GP-S mortars.

The GP-C group of mortars showed no manifestation of efflorescence; in other words, the adjustment of the chemistry in the dosage of the compound activator with the addition of Na_2_SiO_3_ was sufficient to inhibit efflorescence as it improved the interaction of the matrix with the aggregates and reduced the porosity. This result agrees with the study of Longhi et al. [27] since they also observed that the soluble silicate favors the reduction of efflorescence, as it reduces porosity and permeability, as observed in item 3.7.

The molar ratio of Na_2_O/Al_2_O_3_ is a relevant parameter that should be considered in efflorescence control [54]. In this research, the increase in the molar ratio Na_2_O/Al_2_O_3_ was from 0.45 in group S to 0.72 in group C, which was due to the incorporation of Na_2_O contained in the Na_2_SiO_3_ solution. This is another factor that may have contributed to group C geopolymers performing better.

Figure 11a shows in the diffractogram of the efflorescence collected on the surface of the GP-S group mortars, the peak of the hydrated sodium bicarbonate Na_3_ H(CO_3_)_2_^.^2H_2_O and sodium carbonate Na_2_CO_3_ and the micrograph of the efflorescence in Figure 11b.

The crystals that predominate on the surface of the samples originate from the residual alkalis that did not react in the geopolymerization and migrated by leaching and were transported through the porosity of the mortars, as observed in the physical and morphological characteristics of this group. In the morphology shown in Figure 11b, the predominant efflorescence is formed by trona crystals, which are composed of Na_3_ H(CO_3_) _2_^.^2H_2_O (hydrated sodium bicarbonate).

## 4. Conclusions

In this research, we studied the influence of the replacement of sand with plastic waste in geopolymer mortar with the use of two types of alkaline activators. The analyses were performed by comparing the compressive strength, the microstructure, and efflorescence between the two groups of mortar proposed.

It is understood that, when proposing an alternative material different from the conventional one, it is necessary to investigate which effects can contribute satisfactorily or not to the performance of the product obtained, which, for this study, was the mortar.

This study, in addition to being a proposal for an alternative to the use of plastic waste in cementitious materials, provides benefits through the suggested method in order to understand the behavior of raw materials after synthesis and how these are related to plastic waste and types of activators. Based on the analysis of the results, the following conclusions can be made:

Via the TGA and DTA analyses, it was possible to delimit the kaolin calcination temperature at 650 0C to obtain metakaolinite and, via the XRF, identify the total of 92.21% of aluminosilicates with the potential to produce reactive metakaolinite, considering that the XRD using the Rietveld refinement already confirmed content of 90.71% of kaolin. In addition, the HI calculation for kaolin was 1.44, thus indicating that it is a highly ordered structure.

The compressive strength varied between the mortars of each group at each substitution level. For the GP-S group, the resistance of 14.78, 12.47, and 7.71 MPa decreased proportionally according to the increase in plastic waste. The same effect occurred for the mortars of Group GP-C with 35.19, 22.69, and 10.68 MPa. These results were influenced by the porosity of the mortars, which causes defects and contributes to a drop in strength. The most expressive variation in porosity and consequent water absorption was for the GP-S group for the GP0-S, with 64.29 when compared to the GP-C Group for the GP100-S with 42.11. It was found that the determining factor for this is related to the type of activator, considering that the aggregates had a better interaction with the matrix when activated with a compound activator.

Through the morphological evaluation of the mortars using SEM, it was evident that the compound activator in the GP-C group contributed to a better packaging of the aggregates in the matrix, thus reducing the porosity and benefiting the formation of the N-A-S-H gel. A new phase was observed in the XRD of the mortars, i.e., zeolite. This phase is common in NaOH-activated geopolymers and tends to form a more porous matrix, which is different from what was observed in the GP-C group activated with NaOH and Na_2_SiO_3_ and had fewer pores. Via visual analysis, SEM and XRD, the GP-S mortar formed aggressive efflorescence that culminated in the degradation of the material, which was a result of the high porosity and the type of activator. On the other hand, the compound activator was able to eliminate efflorescence.

This work proves that it is possible to obtain a geopolymer mortar by replacing sand with plastic waste, which has the potential for application in construction materials in diversified situations that do not require high strength.

## Figures and Tables

**Figure 1 materials-15-08626-f001:**
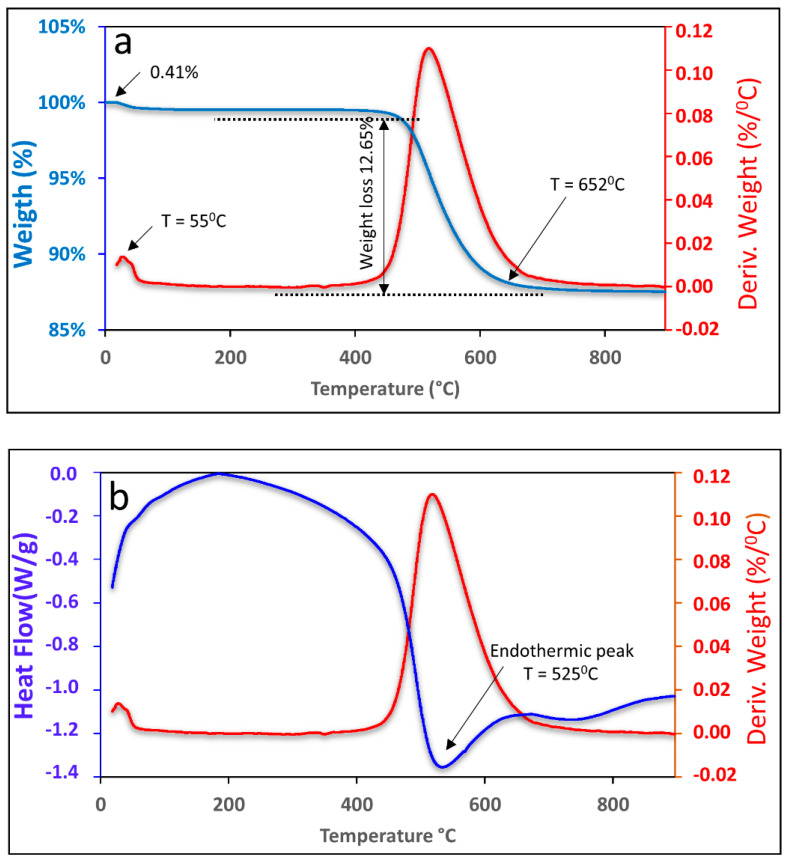
TGA (**a**) and DTA (**b**) curves of kaolin.

**Figure 2 materials-15-08626-f002:**
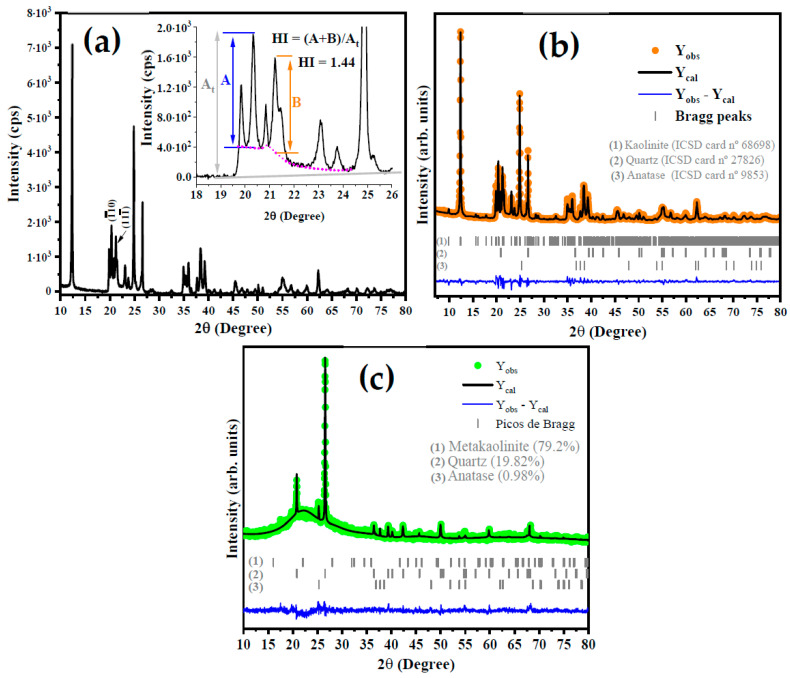
Diffractogram of kaolin with HI calculation model (**a**) and structural Rietveld refinement plot for kaolinite (**b**) and metakaolinite (**c**).

**Figure 3 materials-15-08626-f003:**
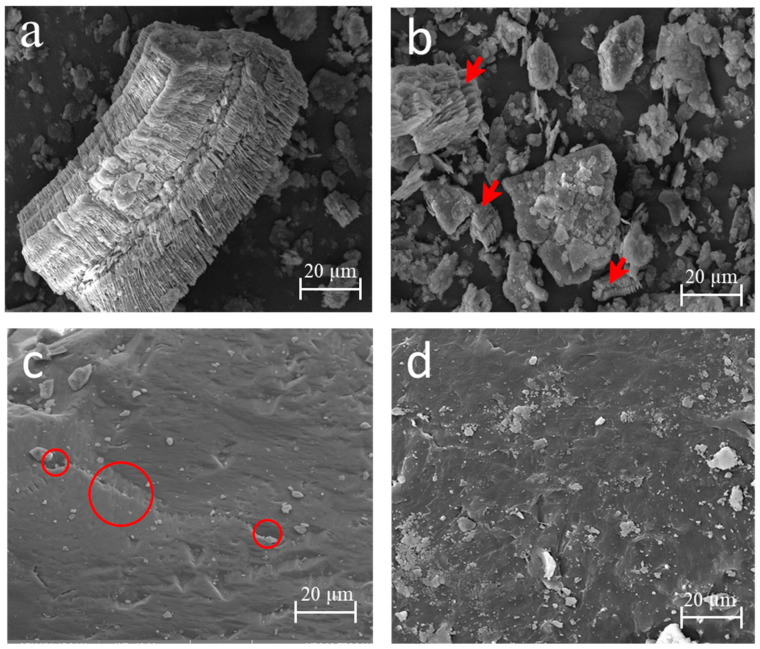
Micrograph image of the raw materials: kaolinite (**a**), metakaolinite with kaolinite destruction indicated by red arrows (**b**), sand with its porosity indicated by red circles (**c**), and plastic waste (**d**).

**Figure 4 materials-15-08626-f004:**
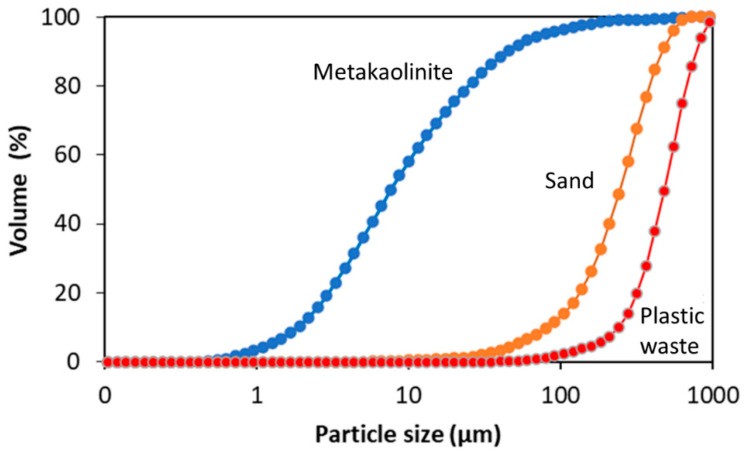
Overlap graph of granulometry curves in the metakaolinite, sand, and plastic waste.

**Figure 5 materials-15-08626-f005:**
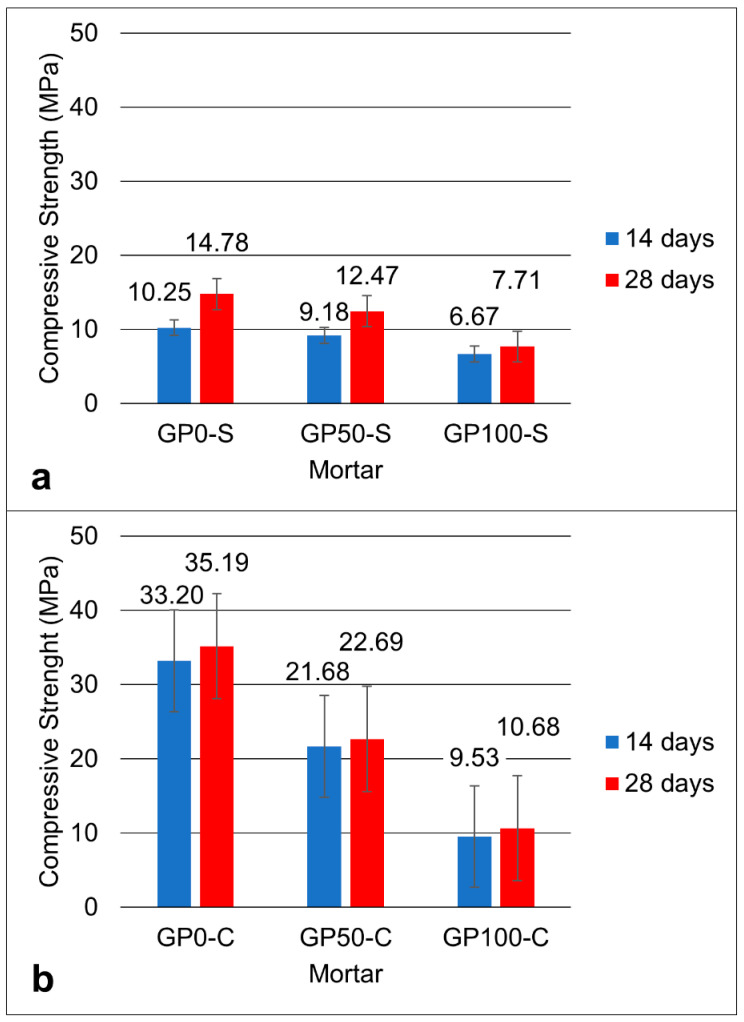
Graph of the compressive strength of the GP-S (**a**) and GP-C (**b**) mortars.

**Figure 6 materials-15-08626-f006:**
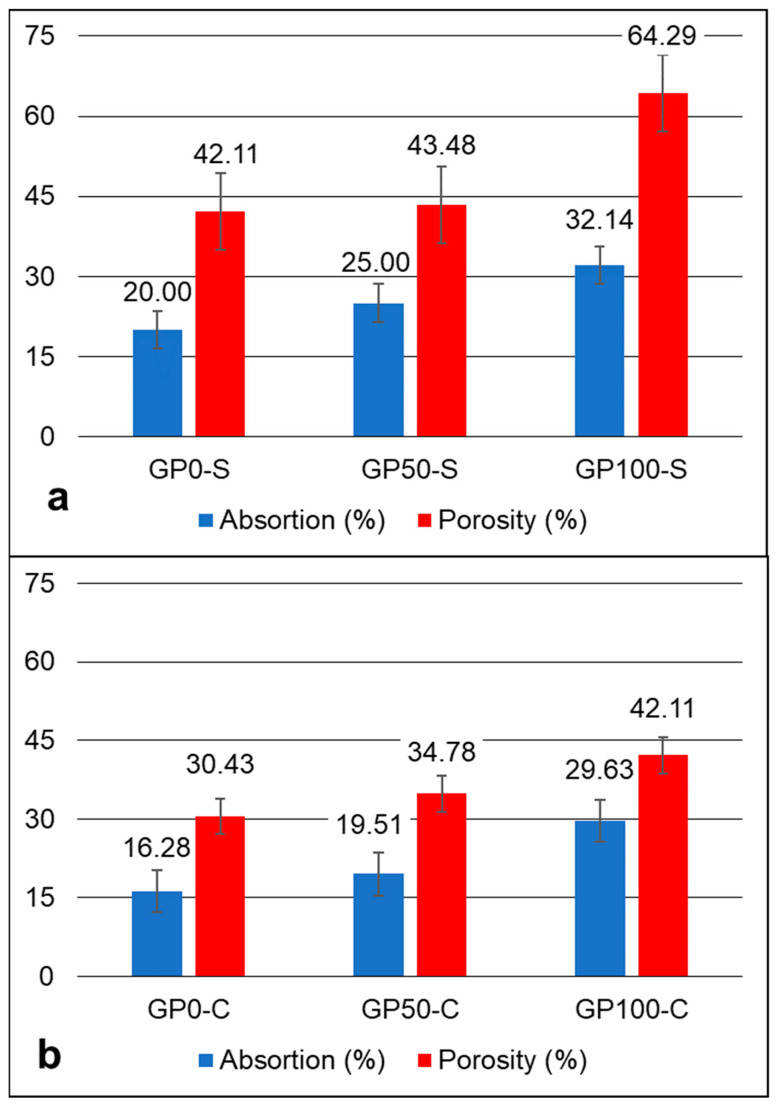
Absorption and porosity graph of the GP-S (**a**) and GP-C (**b**) mortars.

**Figure 7 materials-15-08626-f007:**
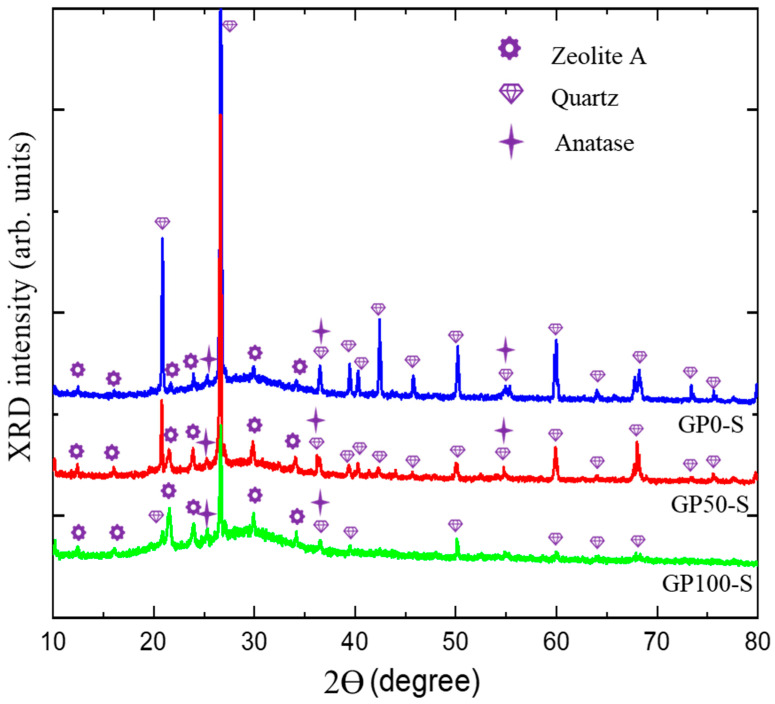
Diffractogram of GP-S group mortars.

**Figure 8 materials-15-08626-f008:**
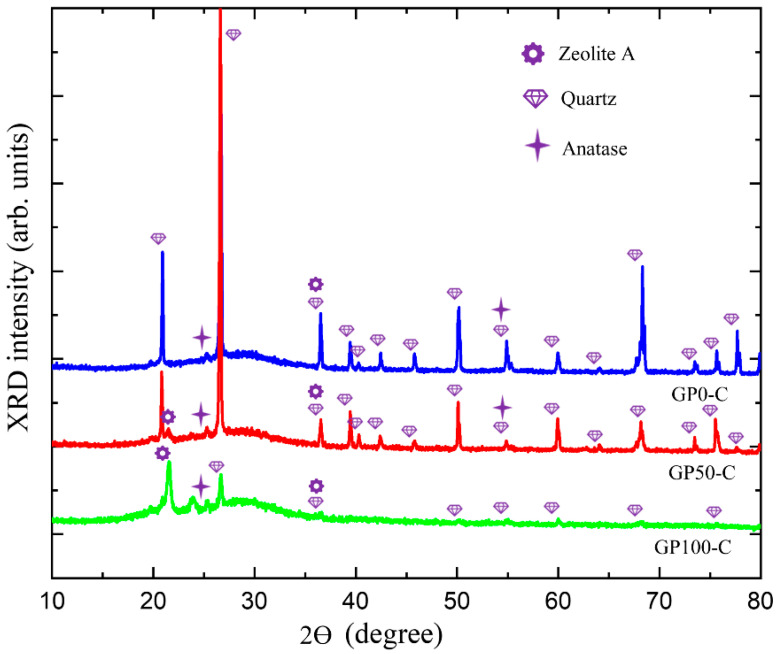
Diffractogram of the GP-C group mortars.

**Figure 9 materials-15-08626-f009:**
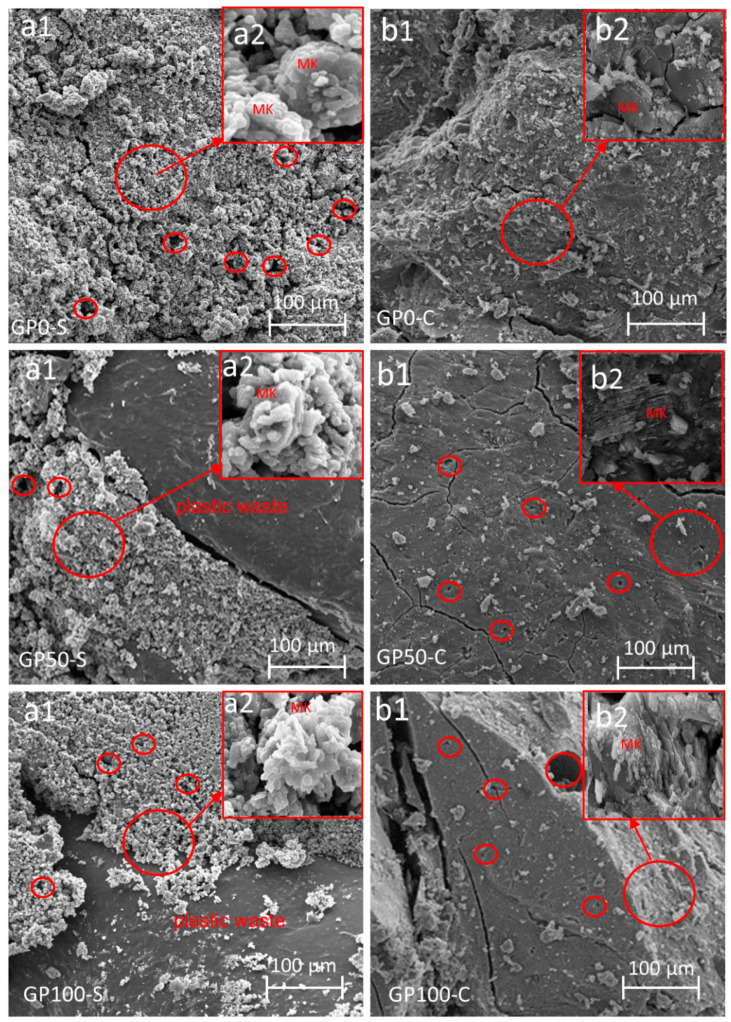
Morphology of the mortars GP0-S (**a1**,**a2**), GP50-S (**a1**,**a2**), GP100-S (**a1**,**a2**), GP0-C (**b1**,**b2**), GP50-C (**b1**,**b2**), and GP100-C (**b1**,**b2**) showing the enlarged area of unreacted material. The areas in the circles show the pores present in the sample.

**Figure 10 materials-15-08626-f010:**
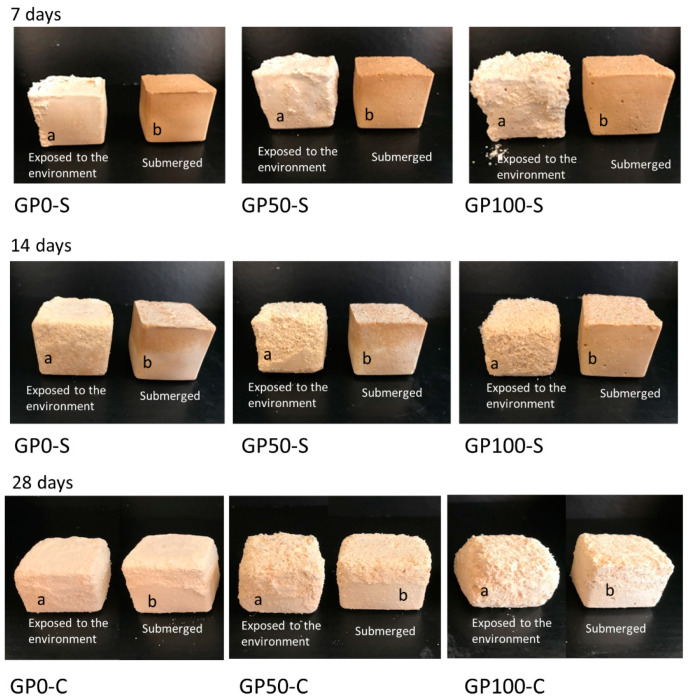
Photo of efflorescence formation with material degradation at 7, 14, and 28 days for samples exposed to the environment (**a**) and submerged (**b**).

**Figure 11 materials-15-08626-f011:**
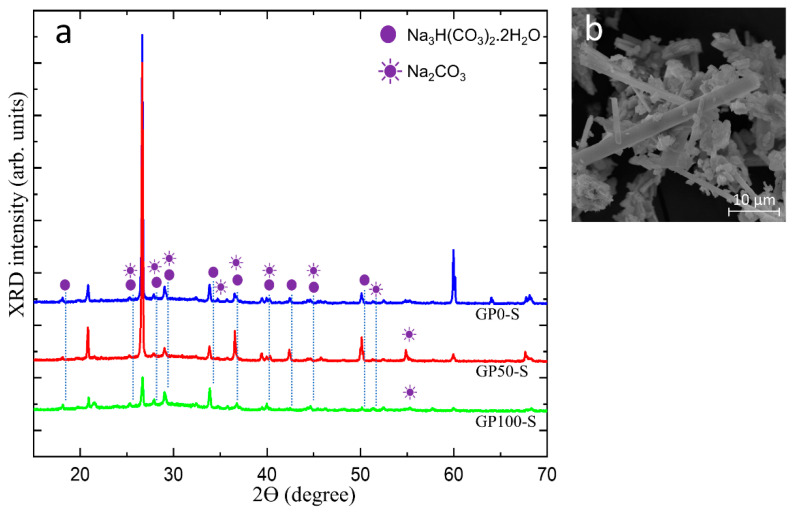
Diffractogram of the efflorescence in group S mortars (**a**) and micrograph of the trona crystals (**b**).

**Table 1 materials-15-08626-t001:** Dosage of materials for the production of the mortars GP-S and GP-C with 0, 50, and 100% replacement levels of sand with plastic waste.

Samples	Metakaolinite (g)	Plastic Waste (g)	Sand (g)	NaOH (g)	Na_2_SiO_3_ (g)
GP0-S	100	0	100	80	0
GP50-S	100	18.4	50	80	0
GP100-S	100	36.9	0	80	0
GP0-C	100	0	100	40	50
GP50-C	100	18.4	50	40	50
GP100-C	100	36.9	0	40	50

**Table 2 materials-15-08626-t002:** Chemical composition of raw materials (Wt%) using the XRF technique.

Oxide	SiO_2_	Al_2_O_3_	P_2_O_5_	K_2_O	CaO	TiO_2_	ZnO	Ag	Fe_2_O_3_	Others
Metakaolinite	51.55	40.66	<0.05	0.33	<0.01	1.14	0	0	0.89	0.50
Sand	96.40	1.07	0.93	0.60	0.44	0.13	0	0	0.21	0.23
Plastic waste	27.64	0	36.36	0	16.48	11.17	1.89	3.46	2.51	0.49

**Table 3 materials-15-08626-t003:** Physical characteristics of the raw materials.

Material	Density (g/cm^3^)	Surface Area (m^2^/g)	d10 (µm)	d50 (µm)	d60 (µm)	d90 (µm)
Metakaolinite	2.68	1.38	1.86	7.65	10.57	44.69
Sand	2.60	0.041	81.77	244.56	282.73	464.13
Plastic waste	0.94	0.016	241.31	480.30	532.35	773.82

## Data Availability

Not applicable.

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
