# Peer review of "Study of Metakaolinite Geopolymeric Mortar with Plastic Waste Replacing the Sand: Effects on the Mechanical Properties, Microstructure, and Efflorescence"

_materials, 2022, doi:10.3390/ma15238626_

Round 1

Reviewer 1 Report

I have found the conducted research very interesting, but something remained highly unclear to comprehend and must be considered to increase the quality of the paper for publication; therefore, I kindly ask authors to prepare a response letter point-by-point rebuttal and must be subjected to the manuscript as well, considering the following comments with sufficient explanations.

1)     What is exactly LPDE in table 1 or other parts of manuscript? Is it LPDE or LDPE plastic waste? It must be clear for the readers and must be explained the difference between them in the methodology.

2)     According to the following literatures (DOI: 10.1007/s42860-020-00082-w), there is not any dehydration at higher temperate of 150 °C, but dehydroxylation takes place above 150 °C, especially in the process of metakaolinite fabrication because of hydroxyl group (OH) is released by forming a water molecule. Therefore, line 256, 257, and 258 must be modified.  It is also suggested to refer your work to some promising studies as this study has done numerical and experimental observations. Their results should also be reported in the introduction.

3)     According to above literature, did you also distinguish the percentage of types of kaolinite and disordered kaolinite contributions and finally their dehydroxylation (DHX) productions of metakaolinite and metadiskaolinite by XRD measurements?

4)     Is there any reason that you have not used KOH, sodium silicate and potassium silicate, as an activator? Please explain it in the methodology what is the main reason for not using these activators? What is the ratio of Si:Al (silico aluminates) after geopolymerization? The reason must be added to the methodology.

5)     In the caption of table 1 must be noticed that the abbreviation of S or C (for two different samples) represents as ….? All the samples must be explained in the captions to be clear for the readers.

6)     As the usage of plastic waste reduces the compressive strength due to the increase in porosity, what is the main aim of using this approach for this study?

7)     Please make more comprehensive conclusion as in the revised version the following points must be included; materials and methods, the significant of this study, the scope of the effort, the procedures used to execute the work, and the major findings.

Author Response

Dear  Reviewer,

Best regurds.

Ivana Soares.

Reviewer 2 Report

The presented manuscript seems to be interesting for readers of the Materials journal, it is written in a good manner and suits the requirements of the journal. It can be accepted for publication after minor corrections listed below.

- English language of manuscript is acceptable in general. However, it would be much better to improve. Please avoid the unnecessary long sentence. Also, some grammatical and typos mistakes can be observed. For examples: show a wide diffuse halo at an angle, nullified the anchoring effect, trona crystals

- Line 161 in the phrase "agitation speed and time of 2.200 rpm" does it mean rotation speed of 2,200 rpm?

- In line 238 of Longhi et al. [24]. is brought; While Longhi et al. [25]. It is true and should be corrected.

- In Figure 1 (a), the location of the expression T= 652 °C should be moved or deleted.

- Lines 280 and 281, the following sentence is not clear and needs explanation: ”In the compound activator group, the SM was 0.94, because, in the calculation of SiO2, the content of this compound present in the Na2SiO3 solution was considered.”

- In line 381 and 357, the word “Anastase” should be corrected.

- Line 392 of the CO2 formula must be rewritten.

- It is suggested to use (a1, a2) instead of (a, c) in the naming of different parts of the figure in the caption of Figure 9.

- In Figure 10, the sum of the atomic percentages of the elements is not equal to 100, especially for GP100-S and GP100-C.

- The caption of Figure 11 should be more complete and the difference between a and b should be specified in each figure. It is necessary to change the word “dias” to “days”.

- The authors have referred to Figure 11b in line 453 of the text (as observed in Figure 11b.), while it is not clearly indicated in Figure 11.

- The caption of Figure 12 should be corrected and part c explained.

- The following sentence is long and unclear and needs to be rewritten: ”In the morphology shown in Figure 12b and 12c, the predominant efflorescence is formed by trona crystals, which are composed of Na3 H(CO3) 2.2H2O (hydrated sodium bicarbonate) with a non-oriented arrangement and clusters among themselves (1). Its morphology varies between elongated (2) and fibrous (3).”

- In the "Conclusion" section, the authors should present more quantitative data as the main results of the research study rather than just some qualitative data.

- Literature review is not sufficient and authors must review and cite more papers in the field and especially newly published ones. Doing this, review and citing the following refs could be helpful:

[] Ceramics International, 42, 2016, 16738-16743.

[] Materials Research Express. 9, no. 2 (2022): 025011.

Author Response

Dear reviewer,

Best regards.

Ivana Soares

Reviewer 3 Report

The article is about a study of metakaolinite geopolymeric mortar with plastic waste due replacing the sand and its effects on the mechanical properties, microstructure and efflorescence. However, some issues must to be addressed:

  1. Abstract: Please start by expressing the aim of this paper, followed by the rest of the information. Typically, the abstract should provide a broad overview of the entire project, summarize the results, and present the implications of the research or what it adds to its field.
  2. The bibliographic foundation is important and well executed, however some new discussions should be inserted, authors should consider some new works in the literature, such as: DOI 10.3390/ma15010202.
  3. ENTIRE RESEARCH is useless if XRF analysis is not made for raw materials.
  4. The graph with granulometric distribution is compulsory to be presented.
  5. Relations (1), (2) and (3) must to be removed: there are very well known.
  6. Figure 2: please indicate the signification for EVERY peak: which phases were obtained?! The results must to be correlated with XRD analysis.
  7. Figure 10: EDS analysis is useless for this kind of materials …
  8. The results are merely presented, not properly discussed. Please add explanations for the observed changes. Please give an extended discussion on the obtained results and correlate your findings with previous literature studies and prospective applications.
  9. More analysis and interpretation of the results should be added for a clearer understanding of observed experimental phenomena.
  10. The authors must to provide some details about importance of the research and their applicability.
  11. Please rewrite the conclusions in a more quantitative form and enhance the clarity of the conclusion section in order to highlight the results obtained.
  12. General check-up and correction of the English language is suggested. There are still some minor typos and grammatical errors.

The author needs to address the abovementioned points for the betterment of the manuscript.

Author Response

Dear reviewer,

Best resguards.

Ivana Soares

Round 2

Reviewer 1 Report

Dear Editors, 

thanks for accurate investigations and responses and it is accepted in the present form.

Best regards,

Reviewer 2 Report

As authors have performed an adequate revise, the manuscript might be accepted for publication in the journal of Materials.

Reviewer 3 Report

The article is suitable for publication.